# Evaluating Radar Reflector Localisation in Targeted Axillary Dissection in Patients Undergoing Neoadjuvant Systemic Therapy for Node-Positive Early Breast Cancer: A Systematic Review and Pooled Analysis

**DOI:** 10.3390/cancers16071345

**Published:** 2024-03-29

**Authors:** Umar Wazir, Michael J. Michell, Munaser Alamoodi, Kefah Mokbel

**Affiliations:** The London Breast Institute, Princess Grace Hospital, London W1U 5NY, UK; umar.wazir@rcsed.ac.uk (U.W.); mikemichell@aol.com (M.J.M.); malamoodi@kau.edu.sa (M.A.)

**Keywords:** neoadjuvant systemic treatment, breast cancer, targeted axillary dissection, radar reflector localisation, Savi Scout, pCR

## Abstract

**Simple Summary:**

Neoadjuvant therapy, which reduces advanced breast tumours before surgery, is common in medical practice. However, accurately mapping lymph nodes post-therapy is challenging and affects staging accuracy. One solution is marking the affected lymph nodes in the armpit before therapy, then removing the marked nodes during surgery via targeted axillary dissection (TAD), which combines standard sentinel lymph node biopsy (SLNB) with pre-neoadjuvant marked lymph node biopsy (MLNB). Radar reflector localisation (RRL) is a promising technology for locating tumours. Our study found that using RRL in TAD is reliable and accurate. This reduces the need for extensive armpit surgery in responsive patients, minimising surgical complications and improving their quality of life without compromising cancer outcomes.

**Abstract:**

SAVI SCOUT^®^ or radar reflector localisation (RRL) has proven accurate in localising non-palpable breast and axillary lesions, with minimal interference with MRI. Targeted axillary dissection (TAD), combining marked lymph node biopsy (MLNB) and sentinel lymph node biopsy (SLNB), is becoming a standard post-neoadjuvant systemic therapy (NST) for node-positive early breast cancer. Compared to SLNB alone, TAD reduces the false negative rate (FNR) to below 6%, enabling safer axillary surgery de-escalation. This systematic review evaluates RRL’s performance during TAD, assessing localisation and retrieval rates, the concordance between MLNB and SLNB, and the pathological complete response (pCR) in clinically node-positive patients post-NST. Four studies (252 TAD procedures) met the inclusion criteria, with a 99.6% (95% confidence [CI]: 98.9–100) successful localisation rate, 100% retrieval rate, and 81% (95% CI: 76–86) concordance rate between SLNB and MLNB. The average duration from RRL deployment to surgery was 52 days (range:1–202). pCR was observed in 42% (95% CI: 36–48) of cases, with no significant migration or complications reported. Omitting MLNB or SLNB would have under-staged the axilla in 9.7% or 3.4% (*p* = 0.03) of cases, respectively, underscoring the importance of incorporating MLNB in axillary staging post-NST in initially node-positive patients in line with the updated National Comprehensive Cancer Network (NCCN) guidelines. These findings underscore the excellent efficacy of RRL in TAD for NST-treated patients with positive nodes, aiding in accurate axillary pCR identification and the safe omission of axillary dissection in strong responders.

## 1. Introduction

Due to the considerable risks associated with it, complete axillary lymph node dissection (ALND) has been largely replaced by the less invasive sentinel lymph node biopsy (SLNB) as the preferred method for assessing axillary lymph nodes in breast cancer patients with clinically negative nodes who are undergoing initial surgery or after neoadjuvant systemic therapy (NST) [1,2]. 

Recent trials, including AMAROS [3] and ACOSOG Z0011 [4], have demonstrated that, for patients with a positive SLNB, omitting ALND does not affect their overall survival (OS). These findings have prompted discussions about expanding the use of less invasive axillary surgery to include patients with clinically positive lymph nodes (cN1) who respond well to NST. However, studies on SLNB in patients with biopsy-proven positive nodes after NST have shown varying rates of false negatives and identification [5]. In our recent meta-analysis, which included over 3000 patients with node-positive breast cancer, we found a false negative rate (FNR) of 13% after NST, surpassing the desired threshold of 10% [5]. 

The studies included in our meta-analysis were heterogeneous, retrospective, and lacked standardization. The variability in FNRs in this context has been attributed to anatomical changes leading to altered lymphatic drainage, NST-induced fibrosis, fat necrosis, granulation tissue formation, or the characteristics of the tumour itself [6]. Hence, the subsequent logical progression was to investigate the potential integration of a postoperative pathological evaluation of the biopsy-proven lymph node that had been tagged before NST. We showed that incorporating the marked lymph node biopsy (MLNB) alongside SLNB in this context is linked to an acceptably low FNR of 5.18% (95% CI: 3.41–7.54) [7]. 

The combination of MLNB and SLNB after NST for biopsy-proven node-positive breast cancer is known as targeted axillary dissection (TAD). Various approaches, including the deployment of markers such as stainless steel, titanium, or polyglycolic acid clips, as well as carbon or black ink tattooing, are used to mark the biopsy-proven lymph node under ultrasound guidance prior to commencing NST [7]. This marking approach is typically followed by a secondary localisation procedure to facilitate the identification and harvesting of the marked lymph node during axillary staging surgery. Wire-free localisation techniques, such as the SAVI SCOUT^®^ localisation system, also known as Radar Reflector Localisation—RRL (Merit Medical, Aliso Viejo, CA, USA), have emerged as preferred methods for localising the tagged lymph node [7]. RRL provides a radiation-free method with a strong clinical efficacy and exceptional accuracy, particularly in reducing magnetic resonance imaging (MRI) artifacts. This approach involves the insertion of a 12 × 1.6 mm electromagnetic wave reflector under ultrasound guidance, using a sterile 16-gauge introducer needle system. During surgery, the reflector is activated by infrared light from the console probe, reflecting an electromagnetic wave signal back to the detection probe. This continuous feedback guides the surgical excision depth, ensuring a precise depth detection of up to 6 cm from the skin surface [8]. The reflector deployment takes place at the time of biopsy (single-stage localisation) or at a later stage as a secondary localisation procedure of a previously deployed tag. SAVI SCOUT^®^ reflectors have FDA approval for soft tissue implantation for an unlimited pre-operative period. We previously demonstrated that SAVI SCOUT^®^ is an effective and time-efficient alternative to wire-guided localisation (WGL) for non-palpable breast lesions, with excellent physician and patient acceptance [8,9,10]. 

A pooled analysis involving 842 reflectors from eleven studies and our internal data indicated a successful deployment and retrieval rate of 99.64%, each using the Savi Scout^®^ localisation system for nonpalpable breast lesions [10]. Additionally, a smaller analysis across four studies revealed a significant difference in re-excision rates between Savi Scout^®^ and wire-guided localisation (12.9% vs. 21.1%, *p* < 0.01), suggesting Savi Scout^®^ as a safe and effective alternative to wire-guided localisation, potentially minimising the need for re-excision procedures and enabling flexible scheduling by separating radiology and surgical interventions in the management of nonpalpable breast lesions requiring excision [10].

It should be noted that the new multi-dimensional and multidirectional SCOUT^®^ MD™ with its variably shaped reflectors was recently approved by the FDA (February 2024).

Several studies have investigated the use of RRL in TAD, and a systematic review and pooled analysis aim to evaluate RRL’s clinical performance during TAD (MLNB plus SLNB), assessing successful localisation and retrieval rates, the concordance between MLNB and SLNB, and the incidence of pathological complete response (pCR) in clinically node-positive patients undergoing NST.

## 2. Materials and Methods

### 2.1. Literature Search

The study was approved by the multidisciplinary breast cancer board of the London Breast Institute.

The literature was reviewed by searches of the PubMed and Google Scholar databases up to February 2024. The following keywords were used: 

[[SAVI SCOUT] OR [radar reflector localisation]] AND 

[[targeted axillary dissection] OR [TAD]] 

[breast cancer] 

[neoadjuvant].

In addition, bibliographies were searched for further studies for potential inclusion and the corresponding authors of the relevant publications were contacted to clarify certain aspects of their data where it was deemed necessary. 

A *p*-value of 0.05 was accepted as the threshold of statistical significance.

The systematic review followed the recommendations of the Preferred Reporting Items for Systematic Reviews and Meta-Analyses (PRISMA). The protocol has not been registered.

### 2.2. Inclusion and Exclusion Criteria

The studies identified in the literature review were assessed according to the following inclusion and exclusion criteria.

#### 2.2.1. Inclusion Criteria

Studies were included if they met the following criteria:Both retrospective and prospective cohort studies.Studies studying the role of RRL in TAD where patients underwent NST.Following data endpoints available: successful localisation and retrieval rate; SLNB-MLNB concordance rate; pCR; and migration rate.

#### 2.2.2. Exclusion Criteria

Studies meeting the following criteria were excluded:Manuscript not available in English.Including non-human subjects.Non-peer reviewed studies.Ten or less eligible cases.Conference proceedings and published abstracts.

## 3. Results

### 3.1. Literature Search Results

The search revealed 139 articles, 4 of which met the inclusion criteria, spanning 252 patients (Table 1) [11,12,13,14]. After inspecting all the articles retrieved using the relevant keywords and search terms, six articles were initially identified. However, upon further analysis, two articles were deemed ineligible due to having less than 10 cases of RRL-guided TAD [15,16]. As a result, four articles met the criteria for analysis (Figure 1).

### 3.2. Pool Analysis

Four studies involving 252 patients met the inclusion criteria. The pooled average age was 54 years (range: 20–91). The pooled analysis revealed a 99.6% (251/252) [95% confidence interval (CI) 98.8–1.0] successful localisation rate, 100% retrieval rate, and 81% (181/224) [95% CI 76–86) concordance rate between SLNB and MLNB. pCR was observed in 42.8% (107/252) [95% CI 36–49] of cases, with no migration or procedure-specific complications reported. The successful deployment rate was also 100%, however, the deployment procedure was repeated in one patient due to an initial deployment 2.5 cm away from the target lymph node [12]. RRL localisation failed in one case (0.4%). In this case, the radioactive tracer effectively identified the sentinel nodes, and the RRL signal was audible before incision and appears to have been lost during surgical dissection. Therefore, the operating surgeon elected to convert TAD into ALND and both the radar reflector and clip were found in the ALND specimen [14].

The pooled average number of lymph nodes retrieved in the TAD procedure was 3.3 (range: 1–13). MLNB retrieved a maximum of four nodes, while SLNB retrieved nine.

Omitting MLNB or SLNB from TAD would have under-staged the axilla in 9.7% (14/145) [95% CI: 4.8–14) or 3.4% (5/145) [95% CI: 4.8–6.4] of cases, respectively.

The pooled average duration from RRL deployment to surgery was 52 days (range: 1–202 days).

The concordance rate between MLNB and SLNB was 81% (95% CI 76–86%). Compared with the final surgical pathology of TAD, the FNR of the MLNB-based axillary staging was significantly lower than that of the SLNB-based staging (3.4% versus 9.7%; χ^2^ = 4.4938, *p* = 0.034, significant at *p* < 0.05).

## 4. Discussion

Currently, more than 20% of patients diagnosed with early breast cancer undergo NST, a percentage that has steadily risen over time [17]. Initially, NST was primarily utilised to downstage locally advanced tumours, facilitating breast-conserving surgery (BCS). However, its scope has expanded to encompass additional objectives such as in vivo drug sensitivity testing and providing crucial prognostic insights to tailor adjuvant treatments for residual disease. Studies have demonstrated that administering adjuvant systemic therapy to patients with residual disease post-NST, particularly in triple-negative breast cancer (TNBC) or human epidermal growth factor receptor 2 (HER2)-positive breast cancer, significantly enhances OS [18,19]. Moreover, advancements in NST protocols have led to increased rates of pCR. Examples of these refinements include incorporating carboplatin for TNBC [20], combining pertuzumab with trastuzumab for HER2-positive breast cancer [21], and more recently, integrating immunotherapy in the form of programmed cell death protein 1 (PD-1) or ligand PD-L1 inhibitors into treatment regimens for early TNBC [22]. These advances have expanded the indications for TAD. While RRL has been extensively studied for localising occult breast lesions [8,9,10], its potential role in localising axillary lymph nodes in the context of TAD is still evolving.

### 4.1. Performance of RRL in TAD

Our study offers a comprehensive analysis of all peer-reviewed publications, spanning 262 procedures, filling a significant gap in the literature regarding the efficacy of RRL in guiding TAD. Our analysis unveiled highly successful deployment, localisation, and retrieval rates nearing 100%, showcasing an excellent clinical performance for RRL in TAD. In the single case of failed localisation, the RRL reflector signal was audible prior to surgical incision and was subsequently lost during axillary dissection [14]. Inactivation of the RRL reflector by electrocautery or mechanical injury could explain this observation. Inadvertent inactivation of the SAVI SCOUT^®^ by electrocautery is rare due to the modern design, which includes additional capacitors to boost its resistance. If such an event occurs, it indicates direct contact between the electrocautery tip and the SAVI SCOUT^®^, allowing the surgeon to visually identify or palpate its edge [8,9,10]. Moreover, the SAVI SCOUT^®^ reflector is easily visible on ultrasonography and, therefore, intraoperative ultrasound could facilitate identification and TAD in this rare scenario. This compares favourably with the successful localisation and retrieval rate of tagged lymph nodes, reported at 90% in our previous pooled analysis involving 1470 marked lymph nodes [7]. Failure to retrieve the target lymph nodes typically results in failure to de-escalate axillary surgery, leading to ALND and its associated morbidity, adversely impacting quality of life.

Achieving 100% target retrieval is essential for accurate axillary staging post-NST. Pathology insights gleaned from a TAD sample analysis could profoundly impact adjuvant treatment planning, particularly with drugs known to significantly enhance OS across all macular subtypes [23]. Additionally, recommendations for adjuvant radiation therapy could also be influenced by TAD surgical pathology. It should be noted that tailoring adjuvant systemic and radiation therapy following NST depends not only on the axillary node status, but also on the presence of residual disease in the breast. The incidence of nodal positivity in the presence of breast pCR was reported to be 30.5% for ER+/HER2-ve, 14.1% for TNBC, and 12.4% for HER2+ve disease [24]. The FNR of TAD compared with ALND was reported to be approximately 5%, which is unlikely to significantly impact OS [7,23]. However, our analysis shows that staging the axilla with SLNB alone could result in a relatively high FNR (9.7%), potentially leading to under-treatment and compromising OS. This is consistent with our previous meta-analysis demonstrating a 13% FNR for SLNB compared with ALND in this clinical setting [5]. Therefore, it is crucial that staging post-NST includes harvesting a biopsy-proven pathological lymph node to accurately assess the response magnitude and likelihood of residual disease. The superiority of TAD over SLNB for staging the axilla after NST in node-positive breast cancer was emphasised in the National Comprehensive Cancer Network (NCCN) 2022 guidelines.

We found that the average duration between the deployment of the electromagnetic reflector and surgery ranged from 1 to 202 days, with a mean of 52 days. This observation underscores the advantages of RRL localisation, such as flexible scheduling and separating the radiological and surgical procedures. Typically, the duration of NST falls between 90 and 180 days. Hence, our findings align with the practice in some centres where target lymph nodes are tagged with clips at diagnosis and the reflector is deployed as a secondary procedure (Figure 2) leading up to the surgery date. In our centre, we typically deploy an electromagnetic reflector during the biopsy of pathological lymph nodes (Figure 3). This ensures a high precision and accurate tagging, thereby saving time and resources by eliminating the need for a second localisation procedure. This practice is supported by the fact that the SAVI SCOUT^®^ reflector is known for its resistance to migration and inactivation over a long period of time [8,9,10,17]. Using Magseed^®^ (Endomagnetics Inc., Cambridge, UK) to facilitate TAD, Barry et al. reported that Magseeds^®^ inserted to localise tagged lymph nodes after the completion of NST were found outside the node in neighbouring axillary tissues in 24.3% of cases, compared to 1.8% when Magseeds^®^ were placed upon the commencement of NST [25]. However, the high incidence (24.3%) of the malpositioning of magnetic seeds reported by Barry et al. after NST has not been corroborated by other investigators.

### 4.2. RRL versus Other Wire-Free Localisation Technologies

The clinical performance of RRL during TAD compares favourably with other wire-free technologies. TAD with ^125^I seed marking before NST achieved a 99.3% identification rate in a Danish cohort of 142 patients [26]. The utilisation of radioactive materials is hindered by intricate regulatory mandates. Furthermore, in certain jurisdictions, the duration for which the seed can remain in the human body is restricted to 5–7 days, thus preventing its deployment at the time of biopsy prior to NST. Preceding insertion, fine-needle aspiration biopsy maybe necessary to pinpoint the intended lymph node for seed insertion. This supplementary procedure necessitates thorough documentation to prevent erroneous deployment in a disease-free negative node [7]. Laws et al. reported a retrieval rate of 91% of the marked lymph node using predominantly Magseed^®^ and radiofrequency identification (RFID) tags (LOCalizer™; Hologic, Santa Carla, CA, USA) in 57 patients [15]. The authors used the RRL only in one case. A notable obstacle hindering the utilisation of RFID tags and Magseeds^®^ for localisation is their tendency to produce signal void artefacts (Figure 4) measuring 2 cm and 4 cm, respectively, in follow-up MRI scans, potentially obstructing the detection of residual disease post-NST when MRI is necessary for treatment response monitoring [27]. Using magnetic seeds to facilitate TAD, prospective data from the ongoing multicentric AXSANA study demonstrated a successful localisation rate of 96.0%. However, in three patients, the seed was removed without detecting any lymphoid tissue on histopathology. The rate of lost markers stood at 1.2% (2 out of 164 magnetic seeds), while MRI assessment was compromised by magnetic seed placement in 15 out of 151 patients (9.9%) [28]. Magseed^®^ also has the added disadvantage of necessitating the removal of all metal equipment from the surgical field during surgical localisation [27]. Additionally, the wide bore of the introducer needle (12-gauge) and susceptibility to migration are further drawbacks associated with LOCalizer™ [29].

Achieving an optimal clinical response is crucial when considering TAD for patients undergoing NST for node-positive breast cancer. pCR rates vary significantly among different breast cancer subtypes, with the highest rates observed in Her2-positive disease and TNBC. Guo et al. reported that, among cN2 patients achieving breast pCR through NST, 78.0% also attained axillary pCR [30]. However, the axillary pCR rate was substantially lower (22%) in patients with ER+/Her2− breast cancer [31]. 

### 4.3. Imaging Modalities for Axillary Monitoring

Exploring non-invasive imaging techniques, such as axillary ultrasonography, MRI, and positron emission tomography with computed tomography (PET/CT) using ^18^F-2-[^18^F]-fluoro-2-deoxy-d-glucose (^18^F-FDG) as a radioactive tracer, has been pursued to accurately evaluate the axillary response in clinically node-positive breast cancer patients undergoing NST, yielding conflicting results. In a recent meta-analysis of 2380 patients, axillary ultrasound demonstrated a pooled sensitivity, specificity, positive predictive value (PPV), and negative predictive value (NPV) of 65%, 69%, 77%, and 50%, respectively [32]. Breast MRI exhibited pooled values of 60%, 76%, 78%, and 58%, while whole-body ^18^F-FDG PET-CT demonstrated 38%, 86%, 78%, and 49%, respectively [32]. Despite these limitations, ultrasound (Figure 5) and MRI (Figure 6) are currently employed in clinical practice to assess responses and guide clinical decisions, with approximate PPV and NPV values of 78% and 50%, respectively [32]. Hybrid PET-MRI scanning may enhance the accuracy of preoperative imaging in predicting pCR [33]. In a pilot study employing this method, De Mooij et al. reported a PPV of 100% and an NPV of 67%. Excluding micrometastases from the node-by-node analysis boosted the NPV to 90% [32].

### 4.4. Oncological Safety of TAD

Global practices in axillary staging methods post-NST for clinically node-positive disease vary, with tumour biology significantly influencing decisions [34]. However, there is a growing body of evidence supporting the oncological safety of reducing axillary treatment in initially node-positive patients responding well to NST. Chun et al. found no statistically significant difference in survival between SLNB and ALND post-NST, establishing SLNB as the gold standard for patients achieving pCR [35]. This was reinforced by a meta-analysis presented at the 2023 San Antonio Breast Cancer Symposium by Rana et al. [36].

The SenTa study was a prospective registry study which compared clinical outcomes in patients who underwent TAD alone and patients who underwent TAD and ALND. A total of 199 patients were included. The data accrued over 43 months suggested that TAD alone in patients with a mostly good clinical response to NST had recurrence rates similar to those seen in patients who had TAD and ALND. Furthermore, the authors found that at least three TAD lymph nodes may confer survival outcomes and recurrence rates similar to TAD with ALND [37]. This study supports the oncologic safety of TAD with a medium-term follow-up. However, its observational design, limited follow-up, and cohort differences are limitations. The suitability of TAD for patients with extensive pre-NST axillary disease remains uncertain. Nevertheless, TAD demonstrates low false negative rates and locoregional failure rates in the medium term. A longer follow-up and new guidelines are needed as TAD becomes standard for more clinically positive breast cancer patients post-NST. The pooled average number of TAD nodes of 3.3 in our analysis is aligned with the SenTa recommendations.

Wu et al. reported on a prospective registry study, where the outcomes were compared between patients undergoing TAD alone or with ALND. After a median follow-up of 36.6 months, 3 nodal recurrences occurred (3/237 with ALND; 0/85 with TAD alone), with 3-year freedom-from-nodal recurrence rates of 100.0% among the TAD-only patients and 98.7% among the ALND patients with axillary pathologic complete response (*p* = 0.29) [38]. Schlafstein et al. reported a retrospective study including patients who had cN1 breast cancer who converted to ypN0 following NST and breast-conserving surgery with SLNB alone and compared the survival in patients who received regional nodal irradiation (RNI) with patients who did not receive RNI. They found no long-standing survival benefit which could be ascribed to RNI and observed that further evidence from the NSABP B-51 trial would have to be awaited, which examined an identical question in a prospective study [39].

Eventually, these findings were confirmed by the five-year data of the NSABP B-51 trial, which was presented at the 2023 San Antonio Breast Cancer Symposium. Specifically, the data from the NSABP B-51 trial confirmed that the de-escalation of axillary treatment did not compromise oncological outcomes in this specific patient subset [40].

Based on the recently presented 5-year data from the NRG Oncology/NSABP B-51/RTOG 1304 study, it seems safe to omit RNI in patients who transition from a cN1 to ypN0 status based on SLNB staging after NST. Since RNI yielded comparable OS outcomes to ALND in the AMAROS trial [3], the NSABP B-51 trial’s findings provide additional evidence favouring the oncological safety of de-escalating axillary surgery in node-positive patients who respond well to NST. This is the only randomised controlled trial to date that has confirmed the oncological safety of axillary surgery de-escalation in patients who achieve pCR [40].

The ongoing TAXIS trial is assessing tailored axillary surgery (TAS), which involves removing biopsied and clipped nodes, sentinel lymph nodes, and palpably suspicious nodes to convert a clinically positive axilla into a negative one. After TAS, patients receive axillary radiotherapy to address any remaining nodal disease. The trial’s primary endpoint is disease-free survival, with secondary endpoints focusing on morbidity and quality of life. With 663 of 1500 patients randomized, completion is expected by 2025. Notably, the trial includes clinically node-positive patients in both NST and upfront surgery settings, exploring surgical de-escalation for high-risk breast cancer patients [41].

### 4.5. Limitations

This study represents the first combined analysis of all published studies investigating the performance of RRL during TAD, encompassing 252 patients. However, the studies analysed lacked standardisation of their protocols regarding the timing of the deployment of the electromagnetic reflector, monitoring responses to NS, the selection criteria for TAD, and the number of lymph nodes to be retrieved. Additionally, two studies included a small sample size of less than 50. None of the studies compared RRL-TAD with other localisation techniques, and none provided data regarding oncological outcomes.

## 5. Conclusions

RRL is a highly reliable and accurate technique for localising pathological axillary lymph nodes in patients undergoing NST for early breast cancer. It facilitates the safe de-escalation of axillary surgery, reducing morbidity and improving quality of life without compromising oncological outcomes.

## Figures and Tables

**Figure 1 cancers-16-01345-f001:**
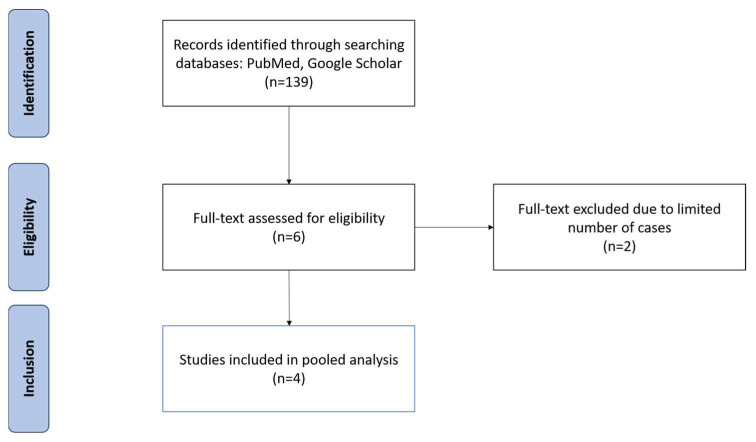
PRISMA flow diagram illustrating the inclusion and exclusion of studies reviewed for this study.

**Figure 2 cancers-16-01345-f002:**
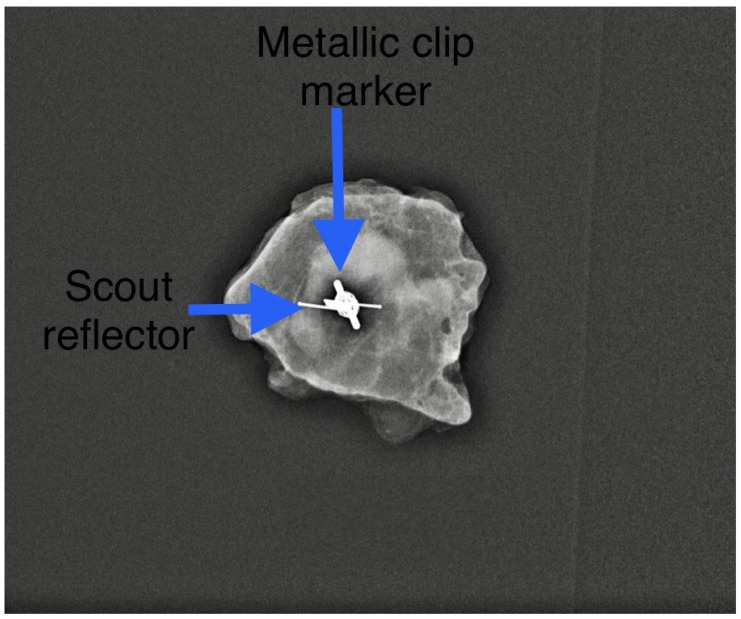
A secondary Scout localisation of a previously tagged lymph node. Reflector indicated by blue arrow. Previously included in our previous publication [7].

**Figure 3 cancers-16-01345-f003:**
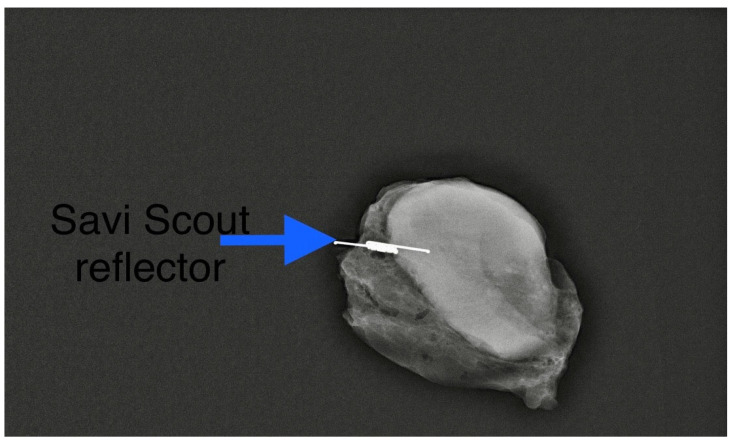
A primary Scout localisation at the time of biopsy. Reflector indicated by blue arrow. Previously included in our previous publication [7].

**Figure 4 cancers-16-01345-f004:**
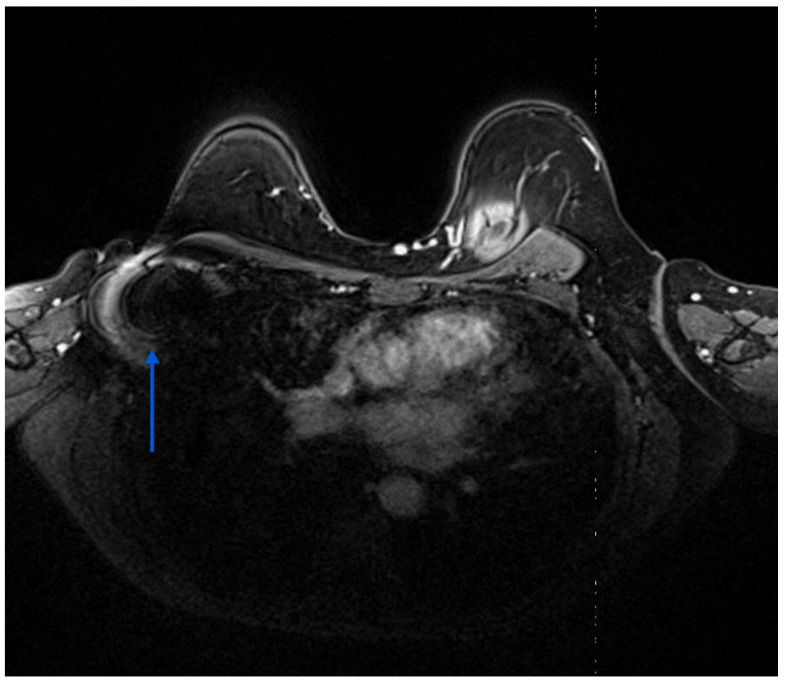
Shows the ferromagnetic artefact (blue arrows) generated by a magnetic seed previously placed in a right axillary lymph node.

**Figure 5 cancers-16-01345-f005:**
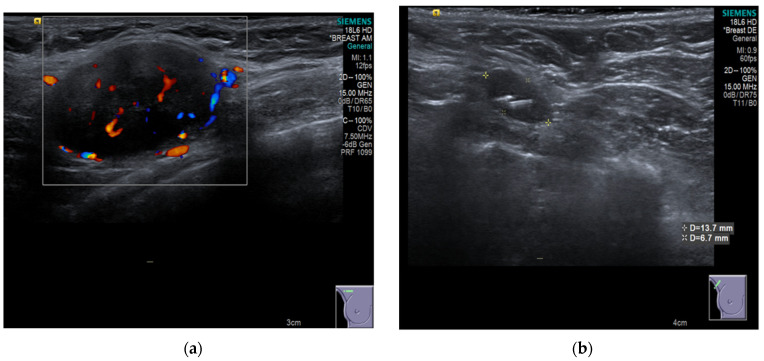
(**a**) Shows the preoperative ultrasound and colour flow Doppler appearance of a pathological lymph node in the right axilla associated with TNBC prior to NST that incorporated carboplatin and Pembrolizumab. (**b**) Shows post NST ultrasound image demonstrating a normal-looking lymph node, containing the easily visible RRL reflector within it. The final surgical pathology of the TAD specimen confirmed axillary pCR [23].

**Figure 6 cancers-16-01345-f006:**
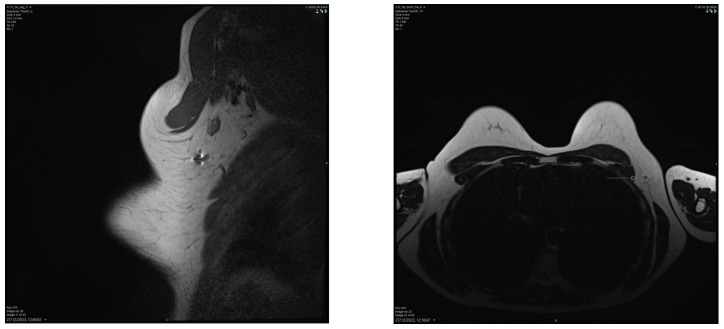
Breast MRI showing a small artefact indicated by an arrow in (4.5 mm in size) in the left axilla related to the Savi Scout reflector within a normal-looking lymph node, suggesting a complete radiological response post-NST. The MRI void signal is significantly smaller than that associated with Magseed demonstrated in Figure 3.

**Table 1 cancers-16-01345-t001:** Pooled analysis of included studies. CI: confidence intervals; MLNB: marked lymph node biopsy; NST: neoadjuvant systemic treatment; and SLNB: sentinel lymph node biopsy. * Median.

Study	Citation	Number of Patients Post- NST	Mean Age in Years	Pathological Complete Response	Retrieval Rate	Localisation Success Rate	Migration Rate	Mean Implantation Duration (Days)	Median Number of Nodes Harvested	SLNB-MLNB Concordance Rate	SLNB Positive/MLNB Negative	SLNB Negative/MLNB Positive
**Baker et al.**	[11]	23	49	10/23	23/23	23/23	0	141	4 (1–8)	22/23		
**Weinfurtner et al.**	[12]	105	57	45/105	105/105	104/105	0	35	-	91/109		3
**Sun et al.**	[13]	45	55 *	20/45	45/45	45/45	0	8	3.5(1.13)	36/45	1	
**Coogan et al.**	[14]	79	51	39/79	79/79	79/79	0	80	3 (0–12)	32/47	4	11
**Total**		252	54 (95% CI: 20–91)	107/252 (42%; 95% CI: 36–49)	252/252 (100%)	251/252 (99.6%; 95% CI: 98.8–100)	0/252 (0%)	52 (95% CI: 1–202)	3.3 (95% CI: 1–3)	181/224 (81%; 95% CI: 76–86)	5/145 (3.4%; 95% CI: 4.8–6.4)	14/145 (9.7%; 95% CI: 4.8–14)

## Data Availability

The datasets generated in this study are publicly available in this open access publication without any restrictions.

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
