# Peer review of "Evaluating Radar Reflector Localisation in Targeted Axillary Dissection in Patients Undergoing Neoadjuvant Systemic Therapy for Node-Positive Early Breast Cancer: A Systematic Review and Pooled Analysis"

_cancers, 2024, doi:10.3390/cancers16071345_

Round 1
Reviewer 1 Report
Comments and Suggestions for Authors
The systematic review and pooled analysis entitled "Evaluating Radar Reflector Localization in Targeted Axillary Dissection in Patients Undergoing Neoadjuvant Systemic Therapy for Node-Positive Early Breast Cancer: A Systematic Review and Pooled Analysis" by Doctor Wazir et al. provides interesting data supporting Radar Reflector Localization (RRL) in Targeted Axillary Dissection (TAD) for breast cancer patients undergoing neoadjuvant systemic therapy (NST).
With a successful localization rate of 99.6% and high concordance between marked lymph node biopsy (MLNB) and sentinel lymph node biopsy (SLNB), RRL demonstrates excellent clinical performance.
The study highlights RRL's role in accurate axillary staging post-NST, crucial for treatment planning.
Author Response
We thank you for your comments and support of our work. We shall endeavour to improve the manuscript as advised.
Reviewer 2 Report
Comments and Suggestions for Authors
The authors present a great systematic review of the localisation of pathological lymph nodes by RRL for targeted axillary dissection. I recommend accepting the article after minor revisions.
Simple summary - it is unclear whether SLNB is part of TAD or not, this should be explained.
Abstract - why "incorporating MLNB into TAD", when MLNB is a usual part of TAD?
Introduction - I recommend moving the paragraph about chemotherapy to the Discussion section.
Methods - The methods should be described in more detail. I highly recommend adding the systematic review flow diagram.
Discussion - "Magseeds were found outside the node in
neighbouring axillary tissue in 24.3% of cases". I recommend mentioning, that this result can be influenced by multiple factors such as the learning curve of the radiologist, because other authors did not mention any results like this.
Author Response
We thank you for your comments and input. We have endeavoured to address your specific comments as follows:
- Simple Summary: We have changed the relavant section to the following to clarify the relation between SLNB and TAD: "One solution is marking affected lymph nodes in the armpit before therapy, then removing marked node during surgery via targeted axillary dissection (TAD), which combines standard sentinel lymph node biopsy (SLNB) with pre- neoadjuvant marked lymph node biopsy (MLNB)."
- Abstract: Your point is take and the phrase has been changed to: "Compared to SLNB alone,..."
- Introduction: The section indicated has been moved to the discussion as advised.
- Methods: We have added a PRISMA chart and have made some additions.
- Discussion: We have added a comment to the indicated paragraph as advised.
Reviewer 3 Report
Comments and Suggestions for Authors
This is a well designed systematic review and meta-analysis . The methodological quality is acceptable and the introduction, methods, results are exhibited correctly. The search strategy of the study selection should be according to PRISMA guidelines and a flowchart should be entered showing the study selection and exclusion. I believe this should be introduced in the text.
Author Response
We welcome your comments and have considered your recommendations. We have included a PRISMA flowchart and have made additions to the methodology section as recommended.